# Hyperspectral Characteristics of an Individual Leaf of Wheat Grown under Nitrogen Gradient

**DOI:** 10.3390/plants10112291

**Published:** 2021-10-25

**Authors:** Jae Gyeong Jung, Ki Eun Song, Sun Hee Hong, Sang In Shim

**Affiliations:** 1Department of Agronomy, Gyeongsang National University, Jinju 52828, Korea; rtyhj5@naver.com (J.G.J.); qjrn00@naver.com (K.E.S.); 2Department of Plant Life Science, Hankyong National University, Anseong 17579, Korea; shhong@hknu.ac.kr; 3Institute of Life Sciences, Gyeongsang National University, Jinju 52828, Korea

**Keywords:** hyperspectral imaging, vegetation index, *Triticum aestivum*, nitrogen gradient

## Abstract

Since the application of hyperspectral technology to agriculture, many scientists have been conducting studies to apply the technology in crop diagnosis. However, due to the properties of optical devices, the reflectances obtained according to the image acquisition conditions are different. Nevertheless, there is no optimized method for minimizing such technical errors in applying hyperspectral imaging. Therefore, this study was conducted to find the appropriate image acquisition conditions that reflect the growth status of wheat grown under different nitrogen fertilization regimes. The experiment plots were comprised of six plots with various N application levels of 145.6 kg N ha^−1^ (N1), 109.2 kg N ha^−1^ (N2), 91.0 kg N ha^−1^ (N3), 72.8 kg N ha^−1^ (N4), 54.6 kg N ha^−1^ (N5), and 36.4 kg N ha^−1^ (N6). Hyperspectral image acquisitions were performed at different shooting angles of 105° and 125° from the surface, and spike, flag leaf, and the second uppermost leaf were divided into five parts from apex to base when analyzing the images. The growth analysis conducted at heading showed that the N6 was 85.6% in the plant height, 44.1% in LAI, and 64.9% in SPAD as compared to N1. The nitrogen content in the leaf decreased by 55.2% compared to N1 and the quantity was 44.9% in N6 compared to N1. Based on the vegetation indices obtained from hyperspectral reflectances at the heading stage, the spike was not suitable for analysis. In the case of the flag leaf and the 2nd uppermost leaf, the vegetation indices from spectral data taken at 105 degrees were more appropriate for acquiring imaging data by clearly dividing the effects of fertilization level. The results of the regional variation in a leaf showed that the region of interest (ROI), which is close to the apex of the flag leaf and the base of the second uppermost leaf, has a high coefficient of determination between the fertilization levels and the vegetation indices, which effectively reflected the status of wheat.

## 1. Introduction

Nitrogen is very important in the growth and photosynthesis of crops. When nitrogen is sufficient, chlorophyll and rubisco proteins increase, the net photosynthetic rate increases, and yield is increased because it has a positive effect on yield components [1]. Makino [2] reported that carbon dioxide assimilation and yield increased with the increase in nitrogen fertilization levels in most wheat varieties. However, excessive nitrogen fertilization does not have a significant effect on the plant height and stem diameter but reduces the culm thickness and dry matter content and reduces the lignin and cellulose content, resulting in increased lodging and reduced yield [3]. Vogeler et al. [4] reported that under continuous cropping conditions, excessive nitrogen fertilization decreased yield and increased nitrogen leaching rapidly. Therefore, it is essential to confirm the optimal nitrogen fertilization level required by crops in terms of crop production and environment.

The typical way to determine the of nitrogen content in plants is the Kjeldahl method, as suggested by J. Kjeldahl in 1883. This method can accurately quantify the nitrogen content of the crops, but it requires a lot of labor and time. Because the Kjeldahl method is a destructive one, it is limited in determining the nitrogen status of the crops immediately in the field. To solve these disadvantages of chemical analysis, portable chlorophyll meters, multispectral cameras, and hyperspectral cameras based on the optical property of leaves have been developed to identify the nitrogen status quickly and easily in crops.

The SPAD meter, a portable device that is widely used in the field, is simple to use, but it is difficult to use on a tiny leaf, and the measured value varies depending on the measuring region in a leaf. The multispectral camera has limited information that can be extracted only by the wavelength of specific bands. On the other hand, the hyperspectral camera is non-destructive, as it measures the light reflected from the plant, and it can measure various spectrums from visible light to infrared light, making it possible to obtain more information easily and quickly. Several researchers have started to apply the hyperspectral technique to nitrogen state analysis in crops since the late 20th century, such as the estimation of nitrogen content [5,6] and the estimation of nitrogen distribution in a leaf [7]. The hyperspectral technique was used in fields to optimize nitrogen fertilization in corn and to evaluate moisture stress [8], nitrogen stress [9], plant disease severity [10], and leaf area index (LAI) and chlorophyll content [11]. Recently, it has been used for predicting chlorophyll content [12], nutritional value evaluation of forage crops [13], leaf nitrogen content estimation [14], and the determination of water content and physiological condition through machine learning [15].

Various vegetation indices used in the determination of the physiological condition of plants are made based on the reflectivity of specific wavelengths extracted from hyperspectral data. NDVI (Normalized Difference Vegetation Index) using red band and near-infrared region reflects the status of plant stress and aging and is highly associated with biomass production [16,17,18]. In addition, Edalat et al. [19] suggested that NDVI could be used for the diagnosis of nitrogen deficiency in the early stage of crop growth. MRE-NDVI (Modified Red-Edge NDVI) is an index that improves the existing vegetation index using red-edge that is sensitive to plant aging and is known to be more effective in estimating chlorophyll content [20,21,22,23]. GNDVI (Green Normalized Difference Vegetation Index) is an index that responds sensitively to chlorophyll changes using the green light band instead of the red light used for NDVI, showing high accuracy in estimating chlorophyll content at the canopy level [23]. The pigment specific simple ratio carotenoids (PSSRc) is a vegetation index associated with carotenoids, an important plant pigment in relation to light absorption and stress response [24]. These vegetation indices allow us to easily diagnose the state of crops by comparing them with well-grown control plants without complex processes such as destructive analysis and modeling.

In this way, hyperspectral analysis has not yet established optimized image acquisition conditions that can affect results in fields despite its many potential applications. Most of the previous studies are based on spectral characteristics of a certain land area using satellites or UAVs, not on an individual plant. The study of hyperspectral analysis of individual leaves for the diagnosis of the condition of crops is insufficient. Considering the reports that the spectral results differ according to angle, target part of a certain organ, and image acquisition [25,26], it is necessary to optimize the imaging condition in hyperspectral analysis for individual organs. This study was conducted to optimize the application of the vegetative indices and the suitable imaging conditions in the hyperspectral analysis of individual leaves for crop growth and physiological analysis under nitrogen gradient conditions.

## 2. Results

### 2.1. Wheat Growth under Various Nitrogen Conditions

The growth of wheat showed a tendency to decrease gradually as the nitrogen fertilization level decreased. In the case of N1, which was 60% more than the recommended rate (100%), the highest value was shown in both the plant height, the chlorophyll content (SPAD), and the leaf area index (LAI). In N6, in which 40% of the recommended rate of N was applied, all growth characteristics examined except Fv/Fm were the lowest. In particular, the difference in the plant height and SPAD greatly reflected the decrease in fertilization level (Figure 1). The nitrogen content in leaves was 2.1% in N1 and N2, but 1.2% in N6 (Figure 2). The N1, the highest fertilization level, showed the greater seed weight, and the N4, N5, and N6 in which nitrogen application rate was lower than recommended rate showed lower yield. In particular, the seed weight in N6 was only 54% compared to N3 and 44.9% compared to N1.

The reflectance of the leaf was higher in N4–N6 than in N1–N3 in visible ranges according to the nitrogen fertilization level, but there was no significant difference in the near-infrared range (Figure 3). The reflectance of the second uppermost leaf was higher in N4–N6, and the reflectance of the NIR region was lower in the N6 with the lowest fertilization level and the highest in N1 with the highest fertilization level. The mean value of reflectance at 500–700 nm in N1 was 80.8% and 75.3% of N6 in the flag leaf and the 2nd uppermost leaf, respectively, showing that the difference of reflectance between N1 and N6 was greater in the 2nd uppermost leaf.

### 2.2. Hyperspectral Reflectances and Vegetation Indices under Various Nitrogen Conditions

The difference in crop growth according to fertilization level was investigated using four vegetation indices of NDVI, MRE-NDVI, GNDVI, and PSSRc calculated from the hyperspectral data. The proportion of standard deviation to average reflectance by light range according to the fertilization level were greater in the 650 nm for spikes, 550 nm and 650 nm for both the flag leaf and the 2nd uppermost leaf (Table 1). The hyperspectral reflectances and vegetation indices were slightly different according to shooting angles. The vegetation indices calculated from the hyperspectral images taken at 125° show a less clear tendency according to nitrogen level. However, the tendency to decrease with the lowering of fertilization level at 105° showed the correlation between vegetation indices and nitrogen fertilization (Figure 4). The second uppermost leaf was more effective than other organs for hyperspectral image shooting in the discrimination of nitrogen fertilization level (Figure 5). By region in a leaf, the apex and base part was more effectively reflected in the difference in the flag leaf and the second uppermost leaf, respectively (Figure 6 and Figure 7). In this study, it was found that the angle reflecting the difference of fertilization level in vegetation indices was 105°. Region of interest (ROI) by organ was apex in the flag leaf, and the base was suitable in the second uppermost leaf. Vegetation indices calculated from hyperspectral reflectances in ROIs of each organ showed differences according to the nitrogen fertilization level. However, the differences were not observed in spikes regardless of image acquisition conditions (Figure 5). It was found that the spike after heading hardly reflects the differences in fertilization level by hyperspectral analysis.

The degree of reflection of the vegetation index on the difference of fertilization level was confirmed based on the determinant coefficient. NDVI, MRE-NDVI, and GNDVI were effective in the base of the 2nd uppermost leaf, and PSSRc was most effective in the apex and distal of the 2nd uppermost leaf showing a higher coefficient of determination (Figure 7). For organs, the flag leaf showed a higher coefficient of determination as it was closing to the apex, and the highest coefficient of determination in the 2nd uppermost leaf was the highest in the base.

The coefficient of determination of the vegetation indices was higher as compared with plant height, SPAD, leaf nitrogen content, and LAI in the second uppermost leaf except for PSSRc (Figure 8). In the case of the plant height, the highest R^2^ was in the base of the second uppermost leaf base, which was similar to the relationship between nitrogen fertilization level and vegetation indices. SPAD showed the highest R^2^ in the second uppermost leaf with vegetation indices except for PSSRc, although it showed differences by vegetation indices. The coefficient of determination between LAI and the vegetation indices except for PSSRc was the highest in the 2nd uppermost leaf, and the vegetation indices using the 2nd uppermost leaf reflecting the variation of fertilization level was suitable for the prediction of growth parameters (Figure 8).

## 3. Discussion

The proportional increase of the plant height, leaf area, chlorophyll content, and stem diameter as the increase of fertilization amount reflects that the nitrogen level is closely related to the plant growth, leaf area index, and chlorophyll content. Conversely, we can estimate the nitrogen state in plants from the chlorophyll content [27]. In similar experiments, the chlorophyll content, leaf nitrogen content, and LAI were significantly increased at 120 kg N ha^−1^ compared to the non-treatment level [28]. Therefore, it was deduced that the nitrogen content in leaves was lowered and chlorophyll was not synthesized sufficiently due to the shortage of available nitrogen in the soil in this experiment. In this experiment, nitrogen fertilization of less than a recommended rate affected growth by reducing the chlorophyll and LAI that closely related to the total photosynthesis.

Hunt et al. [29] reported that the reflectance of visible light and triangular greenness index (TGI) values decreased as the fertilization level increased, and chlorophyll content increased as TGI decreased. There was also a negative correlation between chlorophyll content and reflectance at 500–650 nm [12], and the reflectance at 550 nm was higher in the plant with lower chlorophyll than that with higher chlorophyll [30]. This result shows that the overall reduction in reflectance in the visible range is closely related to the increase in chlorophyll content. The increase in chlorophyll content causes a decrease in reflectance at the visible range because the overall light absorption rate increases due to the high chlorophyll concentration and the reduced relative reflectance due to dark leaf color. In this study, the higher leaf nitrogen content and SPAD value in N1 was confirmed by the reduction of reflectance at visible range through the analysis of the hyperspectral reflectance (Figure 1, Figure 2 and Figure 3), this result implies that sufficient chlorophyll was synthesized for photosynthesis.

That the second uppermost leaf more efficient reflects the nitrogen level than the flag leaf. In the condition of lower nitrogen, senescence progressed faster in the second uppermost leaf than the flag leaf more active in photosynthesis at the heading stage, and the difference in reflectance was greater according to the fertilization level. Therefore, the uppermost second leaf was more suitable for evaluating the fertilization level because the nitrogen in a leaf mobilized to reproductive organs during senescence, which better reflects the effects of nitrogen deficiency. The hyperspectral reflectance decreases in the visible range as the nitrogen fertilization level increases and is known to increase in the NIR region [31,32]. However, in this study, the hyperspectral reflectance of the visible range decreased significantly as the nitrogen fertilization level increased, but the NIR range showed an increase of reflectance that occurs when the nitrogen fertilization level increases only in the 2nd uppermost leaf. In the 2nd uppermost leaf, NIR region reflectance increased slightly in N1, the lowest in N6, and at similar values in other fertilization levels. Therefore, it was less efficient to find out the difference of reflectance at the NIR region according to nitrogen fertilization in the flag leaf than the second uppermost leaf. The reflectance at the visible range is greatly influenced by pigments such as chlorophyll, but the reflectance at the NIR region is influenced by the leaf conditions like thickness and the internal structure [33]. Therefore, when the reflectance at the NIR region is used, it is necessary to check the growth stage, the leaf condition, and the moisture content. This study shows that the 2nd uppermost leaf is more suitable for evaluating the nitrogen level because it can show the chlorophyll content in the leaf using the hyperspectral camera and reflects the difference in reflectance according to the N fertilization rate.

The differences in standard deviations show that the 550 nm and 650nm regions, known as the green peak and red edge region, are relatively more sensitive to differences in nitrogen levels than other ranges. Kong et al. [34] reported that the model performance was poor at the angle where the spikes were photographed more than the leaves among the various shooting angles in fields. In this experiment, the vegetation indices obtained from the spikes did not differ significantly depending on the fertilization level or angle, and it was found that the spikes should be limited in the evaluation of fertilization level compared to the flag leaf and second uppermost leaf. The optimal imaging condition to know the difference among the fertilization levels was to use the apex in the flag leaf and the base in the second uppermost leaf with the shooting of 105°. The low efficiency of spikes in the discrimination of physiological conditions suggested that the hyperspectral reflectance reflecting the nitrogen level can be diluted by the density of spike in the field when the hyperspectral data is obtained using a UAV at a higher position from the crop canopy. Therefore, it is appropriate to take hyperspectral images before heading in case of field unit hyperspectral analysis using UAV or satellite.

In Gramineae crops, the leaf grows from the node, so the apex part is older than the base. The nitrogen content of the shoot, except for the spike, reduces after flowering [35]. The poor reflection of fertilization level in the spike is due to the preferential remobilization of nutrients from vegetative organs to spikes under insufficient nitrogen conditions. The high R^2^ of the apex in the flag leaf is thought to be due to the fact that the apex is relatively more aged old than the base and precisely reflects the growth status of crops. On the other hand, the second uppermost leaf, which appears before the flag leaf and progresses aging first, seems to show high R^2^ because aging occurs faster in the base by translocation of nutrients to sinks organ, unlike the flag leaf, because the base, closer to the stem than the apex, actively loses nutrients like nitrogen when the nutrient remobilization to the spike occurs. Therefore, it was found that the apex in the flag leaf and the base in the 2nd uppermost leaf reflected the fertilization level well, and overall, the base region in the 2nd uppermost leaf was the most appropriate when taking hyperspectral imaging at the individual leaf level. The hyperspectral data obtained at the canopy level also shows relatively high R^2^, but the efficiency was less as compared to that using only the second uppermost leaf due to interference by the spikes. In the case of leaf nitrogen content in the leaf, the high R^2^ was shown in the flag leaf where photosynthesis occurs actively at the heading stage suggesting the flag leaf was more accurate in estimating nitrogen content.

The upper leaves are suitable for the determination of appropriate nitrogen fertilization, the lower leaves are suitable for the diagnosis of nitrogen status [36], the flowering and the initiation of leaf aging are delayed under sufficient nitrogen fertilization conditions [37]. The aging of the second uppermost leaf is faster than the flag leaf because the nitrogen was insufficient during nutrient remobilization. The difference in average reflectance between N1 and N6 at 500–700 nm was larger in the second uppermost leaf than in the flag leaf, indicating that the 2nd uppermost leaf was more suitable for confirming the fertilization level by hyperspectral analysis.

## 4. Materials and Methods

### 4.1. Study Site and Experimental Design

The study site was located at the Gyeongsang National University Experimental Farm (35°08′58″ N 128°05′49″ E) in Jinju, Korea. The soil was silty loam, with nutrient content about 17.4 g kg^−1^ of organic matter, 74 mg kg^−1^ of nitrogen, 80 mg kg^−1^ of available potassium, and 51 mg kg^−1^ of available phosphorus in the topsoil layer. The fertilization ratio of N–P_2_O_5_–K_2_O was 91–74–39 kg ha^−1^, and all amounts of phosphate and potassium fertilizer were applied as basal dressing. The wheat (*Triticum aestivum*) variety used for the experiment was winter type (cv. Jokyeong). After the rotary tillage and fertilization, the drill planting was done with the row spacing of 25 cm at the seeding rate of 140 kg ha^−1^ on 30 October 2018. For weed control, thiobencab herbicide was applied after sowing. Other cultivation management was based on the guideline of the Rural Development Administration. Top-dressing nitrogen was applied two times to vary the nitrogen fertilizer level; the first top-dressing was performed on 20 February and the second was on 5 March (Table 2).

### 4.2. Growth Analysis

Growth and physiological analysis were conducted at the heading stage on 19 April 2019. Plant height and leaf area index were measured for growth analysis and leaf greenness was measured by the portable chlorophyll system (SPAD-502, Minolta, Osaka, Japan) for the second uppermost leaf. Leaf samples were dried at 80 °C for 48 h and powdered leaves using laboratory mill (TUBE-MILL Control, IKA, Wertheim, Germany), and leaf samples were subjected to nitrogen measurement using the elemental analyzer (TruSpec Micro, LECO, St. Joseph, MI, USA). Yield-related characteristics were investigated after harvest. All measurements were conducted with 10 plants.

### 4.3. Analysis of Hyperspectral Properties of Leaves

To investigate the properties of the hyperspectral image in wheat plants according to nitrogen fertilization level, the portable hyperspectral camera (Specim IQ, Specim Ltd., Oulu, Finland) was used. The camera can measure reflectance in a spectral range of 400 to 1000 nm and can measure a total of 204 bands. The spectral resolution was 7 nm and the image resolution was 512 × 512 pixels. The light source was sunlight and the light intensity was about 1250 μmol m^−2^. To minimize the effect of reflection, the background was used by attaching black paper to the acrylic plate, and leaves were attached to the black plate. The camera integration time was set at 1ms, and the time it took to shoot per image was 30 s. The distance between the hyperspectral camera and the subject plate was 30 cm, and the image was obtained around 1:00 p.m. on a clear day. For the calibration of reflectances, a 99% barium sulfate (BaSO_4_) reference plate having 100% reflectance was used in the wavelength range.

The image was calibrated every image acquisition to minimize the error by light condition. To investigate the difference in reflectance according to nitrogen fertilization level and the difference by organs, we took hyperspectral images from 10 plants for spike, flag leaf, and second uppermost leaf per plot. We cut the spikes, the flag leaves, and the top second uppermost leaves and attached them to the black plate. The image acquisition was conducted at two angles, 105° and 125° (based on the direction of the sun), and the differences of reflectance and vegetation indices were confirmed for each organ. The difference of the region of interest (ROI) in an organ was analyzed by dividing the image into five parts: apex, distal, central, proximal, and base, proportional to the area (Figure 9).

The image processing and analysis were performed using the ENVI 5.1 (Excelis Visual Information Solution, Inc. Pearl East Circle Boulder, Co., USA) program. The overall wavelength of 400 to 1000 nm was evaluated for the flag leaf and 2nd uppermost leaf to know differences by N fertilization level. We obtained the coefficients of determination between vegetation indices and fertilization levels and between vegetation indices and growth parameters and then compared coefficients of determination in order to find the region that efficiently reflects the effect of N fertilization.

### 4.4. Statistical Analysis

The growth data were analyzed by using the PROC ANOVA of SAS program and the mean values were compared at *p* < 0.05 level through Duncan’s multiple range test (DMRT).

## 5. Conclusions

The most suitable shooting angle for acquiring hyperspectral images that are used for calculating vegetation indices in an individual leaf showing the differences of growth characteristics of wheat plants subjected to different fertilization levels was 105°, the optimality of the target organs for taking hyperspectral shooting was in the order of the second uppermost leaf > flag leaf > spike in the evaluation of the condition for proper fertilization. Therefore, the most efficient condition of hyperspectral image acquisition in wheat at heading is to use the base of the 2nd uppermost leaf with the shooting angle of 105°.

## Figures and Tables

**Figure 1 plants-10-02291-f001:**
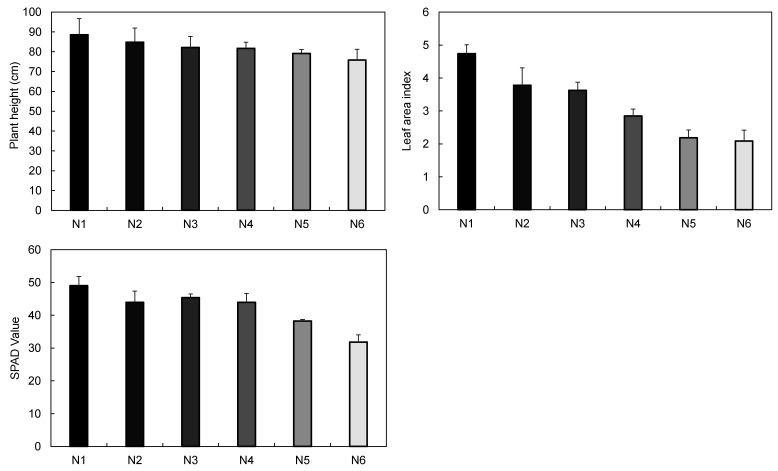
Plant height, leaf area index (LAI), and SPAD value measured on 19 April 2019 in wheat plants with different N top–dressing rates.

**Figure 2 plants-10-02291-f002:**
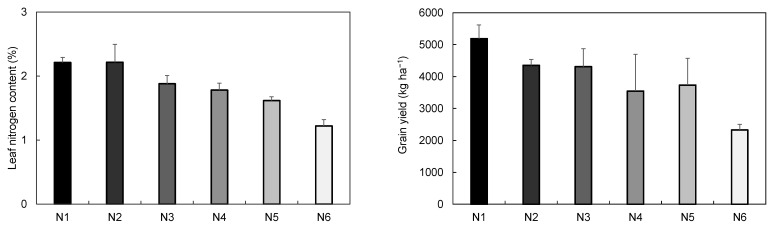
Leaf nitrogen content and seed weight of wheat plants with different N top–dressing rates.

**Figure 3 plants-10-02291-f003:**
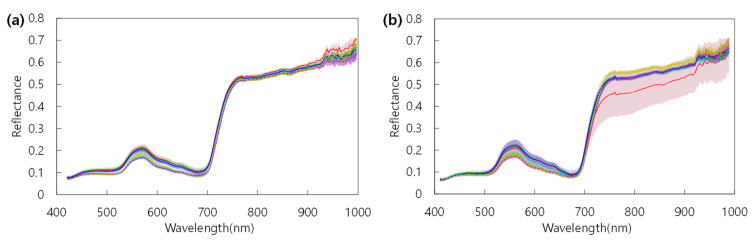
Differences in reflectance by fertilization level in the flag leaf (**a**) and the 2nd uppermost leaf (**b**) at the heading stage. Yellow: N1, magenta: N2, cyan: N3, blue: N4, green: N5, red: N6.

**Figure 4 plants-10-02291-f004:**
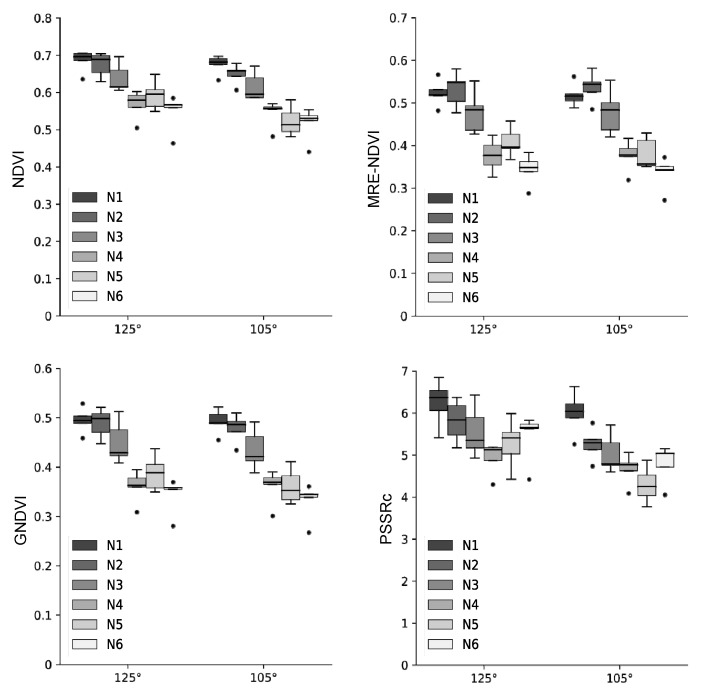
Variation of vegetation indices by the shooting angles for taking the hyperspectral images. Vegetation indices were calculated with spectral reflectance of the base region in the 2nd uppermost leaf. In the box plots, the bottom and top of the box describe the 25th and 75th percentiles, the band near the middle of the box is the 50th percentile (the median); the ends of the whiskers represent the minimum and maximum of the data. The black circle dots indicate outliers.

**Figure 5 plants-10-02291-f005:**
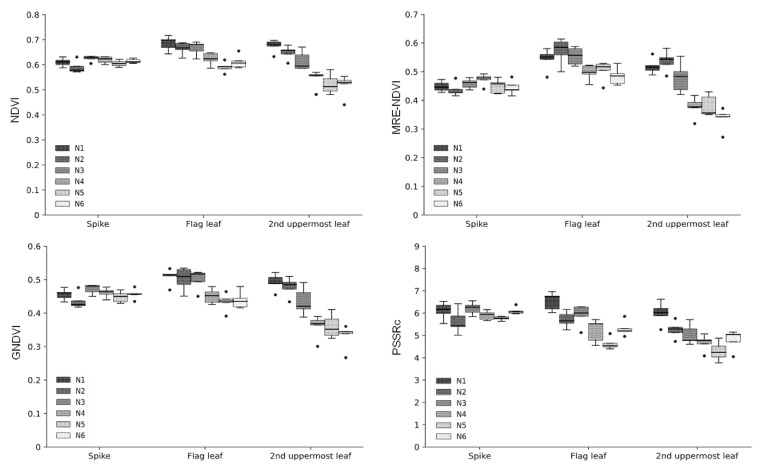
Variation of vegetation indices by the nitrogen levels. Vegetation indices were calculated with the spectral data of the base region in each organ. In the box plots, the bottom and top of the box describe the 25th and 75th percentiles, the band near the middle of the box is the 50th percentile (the median); the ends of the whiskers represent the minimum and maximum of the data. The black circle dots indicate outliers.

**Figure 6 plants-10-02291-f006:**
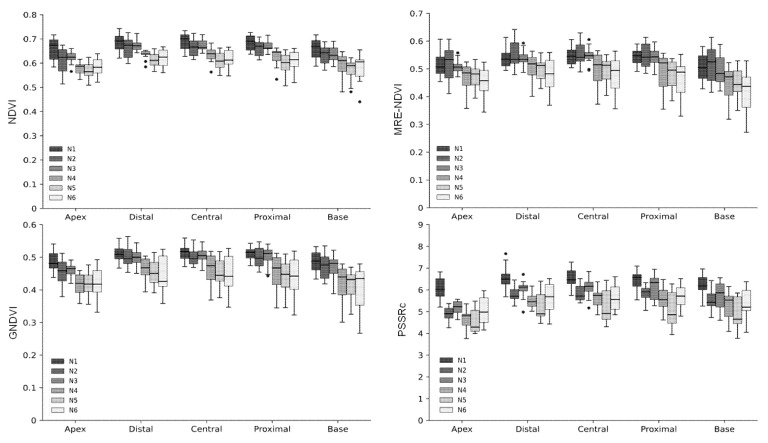
Variation of vegetation indices by the nitrogen levels. Vegetation indices were calculated with the spectral data of five different regions of the second uppermost leaf. In the box plots, the bottom and top of the box describe the 25th and 75th percentiles, the band near the middle of the box is the 50th percentile (the median); the ends of the whiskers represent the minimum and maximum of the data. The black circle dots indicate outliers.

**Figure 7 plants-10-02291-f007:**
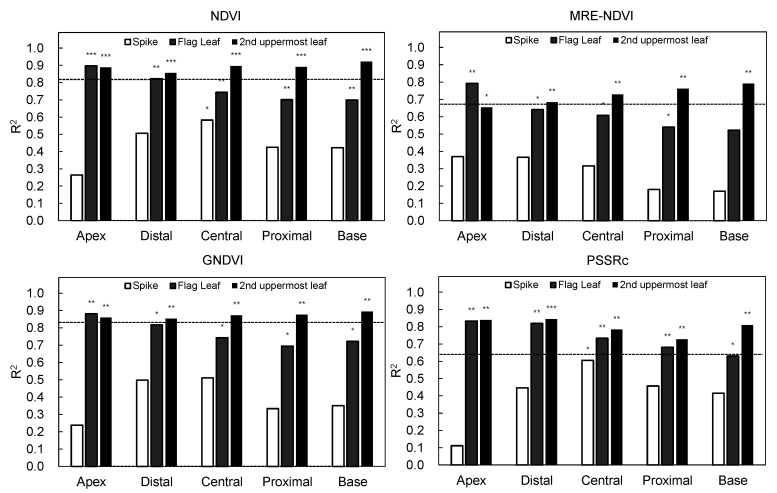
Coefficients of determination by organs between vegetation indices and nitrogen fertilization levels. Horizontal dotted line indicates the R^2^ of canopy level. *, **, and *** mean significant at 0.05, 0.01, 0.001 probability level.

**Figure 8 plants-10-02291-f008:**
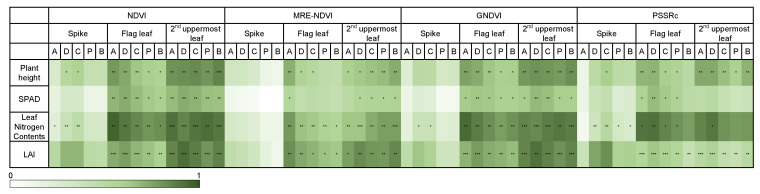
Coefficients of determination by characteristics between vegetation index and SPAD, leaf protein, and leaf area index. A, D, C, P, and B indicate apex, distal, central, proximal, and base, respectively. *, **, and *** mean significant at 0.05, 0.01, 0.001 probability level.

**Figure 9 plants-10-02291-f009:**
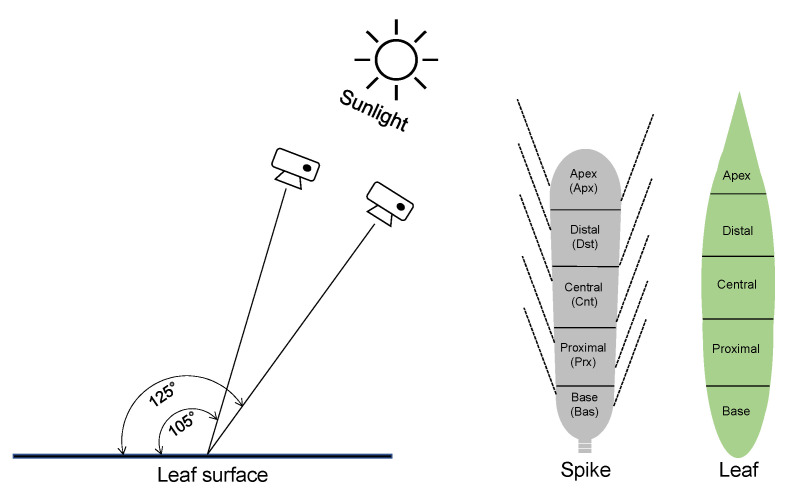
Conditions of hyperspectral image acquisition and separation of regions in leaf and spike for taking hyperspectral shooting.

**Table 1 plants-10-02291-t001:** Standard deviation of the reflectances measured in organs at different wavelengths.

Wavelength (nm)	Standard Deviation (Proportion to Average)		
	Spike	Flag Leaaf	2nd Uppermost Leaf
450	0.004 (4.8%)	0.008 (10.1%)	0.005 (5.3%)
550	0.009 (3.7%)	0.020 (11.4%)	0.027 (14.0%)
650	0.006 (7.5%)	0.014 (16.4%)	0.013 (13.8%)
750	0.030 (4.9%)	0.012 (2.4%)	0.037 (7.4%)
850	0.030 (4.8%)	0.010 (1.8%)	0.032 (6.0%)
950	0.035 (6.3%)	0.023 (3.7%)	0.036 (5.9%)

**Table 2 plants-10-02291-t002:** Nitrogen application regimes in the experiment.

Fertilization	Nitrogen Application Rate (kg N ha^−1^)					
	N1	N2	N3 ^*^	N4	N5	N6
Basal dressing	36.4 (40%)	36.4 (40%)	36.4 (40%)	36.4 (40%)	36.4 (40%)	36.4 (40%)
1st top–dressing	54.6 (60%)	36.4 (40%)	27.3 (30%)	18.2 (20%)	9.1 (10%)	0 (0%)
2nd top–dressing	54.6 (60%)	36.4 (40%)	27.3 (30%)	18.2 (20%)	9.1 (10%)	0 (0%)
Total	145.6 (160%)	109.2 (120%)	91 (100%)	72.8 (80%)	54.6 (60%)	36.4 (40%)

* Recommended rate.

## Data Availability

The data presented in this study are available on request from the corresponding author.

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
