# Peer review of "Hyperspectral Characteristics of an Individual Leaf of Wheat Grown under Nitrogen Gradient"

_plants, 2021, doi:10.3390/plants10112291_

Round 1
Reviewer 1 Report
Nitrogen is one of the most important macronutrients, essential for the synthesis of proteins and nucleic acids. Therefore, it stimulates growth of underground and most of all aboveground parts of plants, giving them an intense green colour thanks to optimal chlorophyll synthesis. For these reasons, nitrogen is one of the most important macroelements for yield and them quality, which is the most important objective for farmers. The submitted manuscript concerns the optimization of hyperspectral methods for measuring nitrogen content in crop plants. A wide range of studies of growth, physiological properties and grain weight of wheat were carried out at varying fertilizer rates, below and above the recommended dose. Nowadays, reflectance spectroscopy is of great interest in a wide variety of environmental studies. Spectroscopic methods are a useful tool for the quantitative and qualitative analysis of soil and plant components. The main reasons for its popularity are its non-invasiveness, the possibility of obtaining information on the chemical nature of the sample tested and, above all, the possibility of making many measurements in a short time. The portable tools, used in the submitted research, allow measurements to be taken directly in the natural growth environment of the plants. Considering that measurements can be subject to technical errors, the authors presented a wide range of tests of many quality parameters on separated leaf and spike fragments. Additionaly, the obtained results were supported by statistical analyses. And in my opinion the aim of the work has been fulfilled in the content of the publication. I consider that the presented research is of great scientific importance, but, above all, it is an important source of information for agricultural practitioners and should therefore be published.
Uwagi:
- Chapter 4 Materials and Methods should be placed as Chapter 2 immediately below the Introduction. It is very difficult to understand the content of a manuscript when the research results are not preceded by a description of the study area and the research methods used.
- Figures 5 and 6 are very difficult to analyse. I propose to include only one, the most characteristics region for spike, flag leaf and 2nd uppermost leaf in the text of the manuscript. The other diagrams I propose to include in the supplementary material.
Uwagi szczegóÅ‚owe:
Line 63, 75, 77, 144: Explain the abbreviation LAI, GNDVI, PSSRc and ROIs in the manuscript text.
Line 109: In the title of Fig 3 there is no explanation for the letters (a) and (b) next to the graphs.
Author Response
Dear Reviewer
Thank you for your review.
I have revised the manuscript as follows.
1. In the journal "PLANTS", "Instructions for Authors" states that materials and methods are placed behind Discussion.
2. I have reconstructed Figure 5 and Figure 6 to be more simple
3. Line 63, 75, 77, 144: Explain the abbreviation LAI, GNDVI, PSSRc and ROIs in the manual text.
- I've modified it.
4. Line 109: In the title of Fig 3 there is no expansion for the letter (a) and (b) next to the graphs.
- I've modified it.
Thank you again for your valuable review.

Reviewer 2 Report
The study seeked to find the appropriate image acquisition conditions that reflect the growth status of wheat grown under six different nitrogen fertilization regimes. The authors conclude that hyperspectral images should use a shooting angle of 105 degrees in their experiment. Also that the region of interest is the second uppermost leaf of wheat.
The methods are properly described and are suitable to conduct the study.
The text is overall well written.
The graphs and tables are descriptive of the results. The statistics used seem fine to treat their data.
Minor comments:
Do not repeat "application of hyperspectral technology" twice in the first sentence of the abstract.
Describe ROI (region of interest) in the abstract
L53: tiny leaves
Author Response
Dear Reviewer
Thank you for your review.
I have revised the manuscript as follows.
1. Do not repeat "application of hyperspectral technology" twice in the first sentence of the abstract.
- I've modified it.
2. Describe ROI (region of interest) in the abstract
- I've modified it.
3. L53: tiny leaves
- I've modified it to “a tiny leaf “
Thank you again for your valuable review.

Reviewer 3 Report
Improvements in remote sensing of plant quality are always welcome. The authors have done a lot of work to improve hyperspectral analysis of wheat leaves, but their work is not adequately presented, unfortunately.
line 20-21 What are N1 and N6? should be explained in the abstract if mentioned.
53 with a tiny
81 in what way?
Fig 2c The bars labelled b and c for DMRT seem to be the same height! How can they be different? In general, it would be good to see SD or SE bars, too.
Fig. 3 use tonne/ ha, not kg
Figs 5 and 6 are terrible. Find some way to present the important data from this mass of tiny lines and bars.
Table 2 please show % SD-- in other words, is the SD increasing as the average increases? that would be normal. You can show this by dividing SD/avg.
Finally-- how many plants were measured each time? the authors did a lot of measurements on separate plant parts, but the total number of plants in each treatment plot and the number of different plants measured is not indicated. This is important for showing that natural variation was taken into account.
Author Response
Dear Reviewer
Thank you for your review.
I have revised the manuscript as follows.
1. line 20-21 What are N1 and N6? should be explained in the abstract if mentioned.
- I've modified it.
2. Line 53 with a tiny
- I've modified it.
3. Line 81 in what way?
- I've modified it.
4. Fig 2c The bars labelled b and c for DMRT seem to be the same height! How can they be different? In general, it would be good to see SD or SE bars, too.
- We have found a mistake in the figure when we adding the letters on bars. Your comment is right, I've modified it. Thank you very much.
5. Fig. 3 use tonne/ ha, not kg\
- Wheat yield is expressed commonly as kg/ha units, and in the journal "Plants" many researchers use it in kg/ha units; please understand.
6. Figs 5 and 6 are terrible. Find some way to present the important data from this mass of tiny lines and bars.
- I have reconstructed Figure 5 and Figure 6 to be more simple
7. Table 2 please show % SD-- in other words, is the SD increasing as the average increases? that would be normal. You can show this by dividing SD/avg.
- Really thank you for your critical comment. I have changed the table according to your comment.
8. How many plants were measured each time?
- We have used 10 plants for each measurement and hyperspectral images. I added the description of the number of plant in the text.
Thank you again for your valuable review.

Round 2
Reviewer 3 Report
Although the authors adopted recommendations that I made, they made no effort to discuss the implications of the changes. There are still changes to be made that will make the article easier to understand.
line 132-- Now that you can see that the SD decreases as a % of the mean as the wavelength increases, your conclusions about which wavelength to use should not be based on size of SD. Please rethink your data.
Renumber the tables, now that Table 1 is at the end, in Materials and methods.
Figs 1 and 2 need SD or SE. The letters of DMRT still seem placed strangely-- there are places where the difference between b and c is less than that between a and b. SD or SE bars will remove those questions.
Figs 4,5,6 -- Each caption of each figure needs a description of the statistics. Sre we looking at SD or SE? What do the asterisks mean? Also, please label each y-axis, rather than have the labels in the caption.
Tables 3 and 4-- what do these mean? They are a jumble of numbers without any statistics to show what is important or statistically significant. Can these somehow be presented graphically? There are much too many numbers to make sense of the data overall.
Author Response
Dear Reviewer
Thank you for your review.
I have checked and revised the manuscript according to your comments.
- After speculating the SD values, I changed the part in the text.
- Renumered the tables and figures.
- In Figures, I added SD bars instead DMRT letters.
- I added the description for box plot figures in the captions.
- I changed Tab. 3, 4 to figures to make sense the data.
Thank you again for your valuable review.
